# A Volumetric Absorptive Microsampling UPLC-MS/MS Method for Simultaneous Quantification of Tacrolimus, Mycophenolic Acid and Creatinine in Whole Blood of Renal Transplant Recipients

**DOI:** 10.3390/pharmaceutics14122547

**Published:** 2022-11-22

**Authors:** Xueqiao Wang, Xinhua Dai, Shiqi Wan, Yu Fan, Lijuan Wu, Huan Xu, Lin Yan, Xingxin Gong, Yamei Li, Yao Luo, Yangjuan Bai, Yi Li

**Affiliations:** 1Department of Laboratory Medicine, West China Hospital, Sichuan University, Chengdu 610041, China; 2The Outpatient Department, West China Hospital, Sichuan University, Chengdu 610041, China; 3Department of Urology, National Clinical Research Center for Geriatrics and Organ Transplantation Center, West China Hospital, Sichuan University, No. 37 Guoxue Xiang, Chengdu 610041, China; 4Department of Urology, West China School of Nursing, Sichuan University, No. 37 Guoxue Xiang, Chengdu 610041, China

**Keywords:** tacrolimus, mycophenolic acid, creatinine, VAMS, UPLC-MS/MS, renal transplantation

## Abstract

(1) Background: Continuous monitoring of tacrolimus (TAC), mycophenolic acid (MPA), and creatinine (Cre) after renal transplantation is vitally important. In this study, we developed a new method based on volumetric absorptive microsampling (VAMS) combined with Ultra Performance Liquid Chromatography–Tandem Mass Spectrometry (UPLC-MS/MS) to simultaneously quantify three analytes including TAC, MPA, and Cre in whole blood. (2) Methods: The VAMS-based UPLC-MS/MS assay used a shared extraction and a single injection to simultaneously quantify the included TAC, MPA, and Cre. Development and validation were carried out following the Food and Drug Administration and European Medicines Agency guidelines for the validation of bioanalytical methods. Moreover, clinical validation for the three analytes was performed in both dried blood spot (DBS) and VAMS. Furthermore, a willingness survey was conducted using the system usability scale (SUS) for renal transplant recipients. (3) Results: The assay was successfully validated for all analytes. No interference, carryover, or matrix effects were observed, and extraction recoveries and process efficiencies were >90.00%. Analysis was unaffected by hematocrit (0.20~0.60, *L*/*L*) and anticoagulants (EDTA-2K). Dried VAMS samples were stable for 7 days at ambient temperature and stable for at least 1 month at −20 °C. During clinical validation, the measured TAC, corrected MPA, and Cre concentrations of VAMS samples met the analytical standards (95.00%, 88.57%, and 92.50%). When more stringent clinical acceptance criteria were set, the results obtained by VAMS (90.00%, 71.43%, and 85.00%) better than DBS (77.50%, 62.86%, and 70.00%). Compared with DBS, the survey found that renal transplant recipients are more inclined to use VAMS. (4) Conclusions: A robust extraction and UPLC-MS/MS analysis method in VAMS tips was developed and fully validated for the simultaneous quantification of TAC, MPA, and Cre concentrations. This method provides analytical support for the one-sample remote monitoring of both immunosuppressive drug concentrations and renal function in allo-renal recipients. Based on the good consistency between this method and the routine detection of venous blood samples and higher patient satisfaction than DBS, we believe that VAMS sampling can be a better alternative to venous whole-blood sampling.

## 1. Introduction

For renal transplant recipients, tacrolimus (TAC) and mycophenolic acid (MPA) are the immunosuppressants commonly used to avoid transplantation rejection [1,2], and creatinine (Cre) is an essential indicator to ascertain renal function stability [3]. Therapeutic drug monitoring (TDM) of TAC and MPA and the regular measurement of Cre rely on hospital visits for venipuncture. Frequent outpatient visits and large volumes (>1 mL) of venous sample collection may cause inconvenience for the patients [4]. In our hospital, transplant recipients come from different places, and most local hospitals cannot perform TDM due to difficult access to facilities. Under the background that our hospital has realized to seek medical advice and issue drug prescriptions online, renal transplant recipients who conduct TDM and renal function monitoring still have to visit the hospital for venipuncture sampling. The result is that they are only regularly reviewing but dedicating plenty of energy and fare to queue and travel. Some recipients have venous samples collected at local hospitals and cold-chain transported to laboratories, which is high-cost and of potential biological risk [5].

Therefore, microsampling techniques via a finger prick are developed as a remote sampling strategy to overcome these shortcomings, which allow domiciliary sampling and support long-distance transportation [6]. Dried blood spot (DBS) sampling is currently the most widely used microsampling technique for remote monitoring, with the advantages of being patient-friendly and cost-effective [7,8,9]. However, it suffers from speckle inhomogeneity and sample volume bias, and the analysis is highly dependent on hematocrit values, resulting in poor recovery and reproducibility of analytical results [5,10,11]. A volumetric absorptive microsampling (VAMS) method is introduced [12], calibrated to encompass a specific volume, which not only takes into account the advantages of DBS but also overcomes the disadvantages of DBS [13,14], and Appendix A shows how it works.

For immunosuppressants and creatinine, viable interchangeability of VAMS samples with intravenous samples has been reported [15,16,17], suggesting that VAMS sampling can be used in transplant recipient care. Compared with DBS studies, there are fewer VAMS-based articles on immunosuppressants and creatinine, especially because most of them only quantify one or more immunosuppressant(s) or monitor creatinine separately. In our study, we developed and analytically validated a method to use VAMS combined with Ultra Performance Liquid Chromatography–Tandem Mass Spectrometry (UPLC-MS/MS) to simultaneously quantify three substances in the blood, including TAC, MPA, and Cre. Moreover, VAMS was clinically validated in renal transplant recipients, and DBS was added for method comparison. Finally, a questionnaire was conducted using the system usability scale (SUS) to assess the usefulness of VAMS and DBS. What we did was made into a schematic diagram, expecting to provide technical support for remote microsampling-based renal transplant recipient monitoring, as shown in Figure 1.

## 2. Materials and Methods

### 2.1. Reagents and Materials

VAMS samplers (Mitra^TM^, 20 μL microsampling device) were supplied by Neoteryx (Torrance, CA, USA). The standards of TAC (97% purity) and MPA (98% purity) were purchased from TRC (Toronto, ON, Canada) and Cre (>99% purity) standards were purchased from TCI (Tokyo, Japan). We used a stable isotope-labeled internal standard in which TAC [13C,2H4] (98% purity) was purchased from Alsachim (Illkirch Graffenstaden, France), MPA-d3 (98% purity) was purchased from TRC (Toronto, ON, Canada), and Cre-d3 (98% purity) was purchased from CIL (Andover, MA, USA).

HPLC-grade methanol, acetonitrile, and formic acid were obtained from Thermo Fisher (St. Louis, MO, USA). Ammonium acetate was obtained from Sigma-Aldrich (St. Louis, MO, USA). Zinc sulfate heptahydrate was obtained from Sinopharm Reagent (Shanghai, China). Deionized water was prepared by a Milli-Q integral ultrapure water machine (Merck Millipore, Germany).

### 2.2. Equipments and Conditions

We used a triple quadrupole LC-MS/MS consisting of an ACQUITY UPLC system in combination with a Xevo TQ-S triple quadrupole mass spectrometer, from Waters (Milford, CT, USA). Method optimization was performed using an Acquity UPLC BEH C18 1.70 μm 2.10 mm × 100 mm analytical column. The autosampler temperature was set at 8 °C and the column oven temperature was set to 50 °C. Mobile phases A and B consisted of 2 mM ammonium acetate and 0.10% formic acid, the former in deionized water and the latter in methanol at a flow rate of 0.30 mL/min. The gradient elution program was set to 2% B for 1 min, then the percentage of B was increased to 50% B in 0.50 min, then increased to 90% B in 1 min, and held 90% B for 1.70 min until 4.20 min, then set B to 2%. The total injector run time was 5 min and the injection volume for a single analysis was 2 μL.

The mass spectrometer generates positive ions in electrospray ionization (ESI) mode and the analysis was performed in multiple reaction monitoring (MRM) mode. Mass spectrometer settings were as follows: the cone voltage was set to 30 V, the desolvation temperature was 550 °C, and the source temperature was 150 °C. Adjust and optimize all precursor ions, product ions, cone voltages, and collision voltages to optimum and retention time, as shown in Table 1.

### 2.3. Calibration Standards and Quality Control Samples

TAC and MPA standards were weighed to prepare 1.00 mg/mL stock solutions with methanol. Cre standard was prepared in ultrapure water at 0.20 mmol/mL. TAC and MPA isotope internal standards were weighed to make a 0.04 mg/mL stock solution with methanol. Cre internal standard stock solution was prepared in ultrapure water at concentrations of 1000.00 nmol/mL. The stock solutions were dispensed into 1.50 mL centrifuge tubes and stored at −80 °C. Working solutions of TAC, MPA, and Cre standards and internal standards were prepared in methanol.

The concentration range of the calibrators were 0, 0.50, 1.00, 2.50, 5.00, 12.50, 25.00, and 50.00 ng/mL for TAC; 0, 0.025, 0.05, 0.10, 0.25, 0.50, and 1.25 μg/mL for MPA; 0, 10.00, 20.00, 50.00, 100.00, 250.00, 500.00, and 1000.00 nmol/mL for Cre. For TAC, 0.50 ng/mL was set as the lower limit of quantitation (LLOQ), 1.50 ng/mL was set as the low concentration quality control (LQC), 20.00 ng/mL was set as the medium concentration quality control (MQC), and 40.00 ng/mL was set as the high concentration quality control (HQC). For MPA and Cre, 0.025 μg/mL and 10.00 nmol/mL were set as LLOQ, 0.075 μg/mL and 30.00 nmol/mL as LQC, 0.50 μg/mL and 400.00 nmol/mL as MQC, and 1.00 μg/mL and 800.00 nmol/mL as HQC. Calibration standards and quality control samples were prepared in EDTA whole blood. Fresh calibration standards and QC samples were prepared for each analysis and discarded after the analysis was complete.

### 2.4. Shared Sample Extraction

The prepared standard and quality control samples containing TAC, MPA, and Cre were added to blank whole blood, where the equilibration time following the addition of analytes was assessed. This setting is considered in that there is a balance time for the standard from adding to evenly distributing to the whole blood until it is stable, including T1 (aspirate immediately after mixing), T2 (aspirate after half an hour at ambient temperature), T3 (aspirate after one hour at ambient temperature), and repeated three times at each specified time point. The VAMS samples were prepared according to the instructions by making the device form a 45-degree angle with the blood sample surface, fixing the absorption tip in the blood sample (incomplete invasion) until it turned red, and then staying for two seconds before taking it out. After drying for three hours, the tip was removed and put into a centrifuge tube with 200.00 μL deionized water added and vortexed at 91× *g* for 10 min (Talboys, Troemner, NJ, USA). Then, 300.00 μL of methanol-water (80:20, *v*/*v*) solution containing 40 mM zinc sulfate heptahydrate and 50.00 μL of the internal standard mixture of the three analytes were added, vortexed at 91× *g* for 3 min, and then centrifuged at 11,269× *g* for 5 min. The supernatant fluid was transferred to a 96-microwell plate (Thermo Fisher, USA), sealed, and submitted for HPLC-MS/MS analysis.

### 2.5. Bioanalytical Validation

Development and validation were carried out following the Food and Drug Administration and European Medicines Agency guidelines for the validation of bioanalytical methods [18,19]. Accuracy was expressed by deviation %, calculated from [(measured concentration − theoretical concentration) × 100]/theoretical concentration, precision was the coefficient of variation, the formula was CV (%) = (standard deviation/mean) × 100. N represented the number of sample preparations.

#### 2.5.1. Standard Curve and Lower Limit of Quantitation

Prepared blank samples, zero concentration samples, and the concentration point samples (in duplicate), as well as three standard curves were evaluated within three days. The LLOQ was investigated by marking the nadir of the curve 10 times in the same analytical run and 10 times between runs.

#### 2.5.2. Selectivity and Carryover Effect

Blank whole-blood samples from six different sources were selected for selectivity evaluation. A blank sample was injected after the upper limit of the quantification sample to assess carryover effect and the experiment of carrying contamination rate was carried out by cross-injecting low and high concentration samples.

#### 2.5.3. Accuracy and Precision

Four concentration levels, including LLOQ, LQC, MQC, and HQC, were prepared, five samples at one concentration level, three batches of analysis were completed within three days, and intra- and interassay analysis was performed to assess accuracy and precision.

#### 2.5.4. Matrix Effect, Extraction Recovery, and Process Efficiency

Six batches of blank matrices from different donors were selected, low and high concentration analytes and internal standards were added after extraction, and pure solutions containing the same concentrations of analytes and internal standards were prepared to study the matrix effect. At the same time, corresponding LQC, MQC, and HQC were prepared to calculate extraction recovery and process efficiency.

#### 2.5.5. Dilution Integrity

Samples above the upper limit of quantitation were prepared by the addition of analytes, which were diluted 2-fold with the calibration range for TAC and Cre with a blank matrix, and the dilution factors for MPA were increased to 5, 10, and 20 to investigate dilution integrity.

#### 2.5.6. Effects of Hematocrit and Anticoagulants

Four whole-blood samples were processed into different hematocrit levels (0.20, 0.30, 0.40, and 0.60, *L*/*L*) to prepare the LQC and HQC to assess the effect of the hematocrit level on the quantification of the three analytes [20]. To observe the effect of anticoagulants on the detection, venous whole-blood-spiked samples without anticoagulant EDTA-2K and venous whole-blood-spiked samples with EDTA-2K were prepared.

#### 2.5.7. Stability

The stability of the extracted LQC and HQC (N = 5) in the autosampler for 24 h was determined. After drying the spiked LQC and HQC samples (N = 3) for three hours, they were individually put in a sealed bag with desiccant to simulate the shipping environment of the dried VAMS samples: 3 days at −20 °C, 23 °C, 37 °C, 4 °C, and 60 °C (simulates extreme shipping conditions), 7 days at −20 °C, 23 °C, 37 °C, 4 °C, and 60 °C, and one month at 23 °C, 4 °C, and −20 °C, testing long-term stability.

### 2.6. Clinical Validation

We plan to randomly collect 40 renal transplant recipients with unlimited transplant time. The study was approved by the Research Ethics Committee of West China Hospital of Sichuan University, China (the approval code is 2017-397), and written informed consent was obtained from all subjects. Sample collection was performed by two trained professional nurses to ensure sample quality. Capillary-filled VAMS samples and DBS samples dripped onto Whatman FTA cards were obtained by finger pricks from the renal transplant recipients. Matched venous blood samples were then collected for quantification of whole blood tacrolimus, plasma mycophenolic acid, and serum creatinine, using laboratory methods certified by the College of American Pathologists (CAP). The concentration of tacrolimus in whole blood was detected by the chemiluminescence microparticle immunoassay (CMIA) method, and mycophenolic acid and creatinine in serum were quantified by enzyme-multiplied immunoassay test (EMIT) method and creatine oxidase method. VAMS and DBS samples were dried for at least three hours and stored at −20 °C until analysis, and the latter were analyzed using the laboratory-established LC-MS/MS method [17,21]. At the same time, we conducted a questionnaire survey (SUS) on the renal transplant recipients who tried VAMS and DBS to investigate their availability.

### 2.7. Statistical Analysis and Computing

Data acquisition and quantification were performed using MassLynx V4.1 SCN905 (Waters Inc.). The calibration curves were constructed using a 1/× weighted regression. Various calculations were performed using an Excel (version 2010, Microsoft, USA) spreadsheet and Medcalc (Ostend, Belgium). Normally distributed data are expressed as mean ± standard deviation, and non-normally distributed data were expressed as medians (interquartile range). To investigate the relationship between fingertip capillary sampling strategy and venous concentration, methods were compared using Passing–Bablok (PB) analysis and generating the correction equation [22]. The agreement between TAC, MPA, and Cre measured and estimated concentrations versus VAMS concentrations were evaluated using PB regression analysis, with 95% confidence intervals (95% CI). The Bland–Altman analysis was used to calculate bias [23]. Whole blood (WB), plasma (PL), and serum (SE) concentrations were predicted from both the VAMS and DBS samples according to the method described [24]. Prediction performance was measured using root mean square prediction error (RMSE), mean percent prediction error (MPPE), and median absolute percent prediction error (MAPE), with acceptance limits set at <15% for MPPE and MAPE. The analysis criterion was set to have an absolute prediction error (APE) < 20% for at least 67% of the paired samples [18,19]. The clinical acceptance limit was set to within ±15% of the concentration deviation of the matched samples [20]. The results of the SUS questionnaire were calculated according to its scoring principle [25]. Specifically, for odd-numbered questions, 1 was subtracted from the score obtained, and for even-numbered questions, 5 was subtracted from the score. The final scores for all questions were added together and multiplied by 2.50 to calculate the devices’ SUS usability score.

## 3. Results

### 3.1. Equilibration Times

Concentrations measured by sampling at different equilibration times (T1, T2, T3) ranged from −4.26% to 12.43% in TAC, −12.40% to 1.59% in MPA, and −9.91% to 7.30% in CRE. The average CV of each analyte was, respectively, 4.30%, 4.10%, and 3.50%, so in the subsequent experiments, we chose to mix well after adding the analyte and then immediately aspirate the sample.

### 3.2. Bioanalytical Validation

#### 3.2.1. Standard Curve and Lower Limit of Quantitation

The calibration curves for TAC, MPA, and Cre were linearly correlated with R^2^ all exceeding 0.99. The maximum deviation of intra- and interassay LLOQ in TAC was 10.00%, −12.00% for MPA, and 12.89% for Cre, and CVs were shown to be 6.81%, 3.73%, and 6.87%, all less than 20%. Figure 2 shows the calibration curves and LLOQ chromatograms for all analytes.

#### 3.2.2. Selective and Carryover Effect

Interference values < 20% LLOQ and <5% IS were observed for the channels of analytes and internal standards in blank matrix from different sources. After injection of ULOQ, no obvious residue was found in the blank sample, and the carrying contamination rates of TAC, MPA, and Cre were −9.95%, −4.35%, and −6.46%, respectively. Representative ion chromatographic peaks in the blank sample and blank sample spiked with the internal standard are shown in Appendix A

#### 3.2.3. Accuracy and Precision

All obtained assay results met the acceptance criteria (Table 2). The overall deviation range for the three-day accuracy of TAC was shown to be −3.03~10.50%, −12.25~−3.50% for MPA, and −5.45~4.93% for Cre. The intra- and interassay coefficients of variation were both <15%.

#### 3.2.4. Matrix Effect

The internal standard normalized matrix factors for low and high concentrations of TAC in six different matrices ranged from 0.91 to 1.13 and 1.00 to 1.03 with CVs of 7.05% and 1.00%. For MPA, the ranges were 0.89~0.98 and 0.95~1.01, and the CVs were 3.09% and 2.06%. For Cre, the ranges were 0.96~1.15 and 0.98~1.08, and the CVs were 6.15% and 3.77%.

#### 3.2.5. Extraction Recovery and Process Efficiency

Extraction recoveries for LQC, MQC, and HQC for the three analytes fluctuated between 95.01% and 105.89%, with process efficiencies ranging from 95.88% to 109.28%. Both recoveries and process efficiencies were obtained between 85% and 115%, which met the acceptance criteria (Table 3).

#### 3.2.6. Dilution Integrity

The average recoveries obtained by diluting the twofold high-concentration TAC (100 ng/mL) and Cre (2000 nmol/mL) were 103.51% and 99.78%, and the CVs were 1.99% and 0.18%, respectively. When the MPA dilution factors were 5, 10, and 20, the average recoveries were 104.60%, 93.71%, and 100.16%, and the CVs were all less than 4%.

#### 3.2.7. Hematocrit Effect and Effects of Anticoagulants

When evaluating the effects of different HCTs, the maximum mean deviations were −3.87% for low concentrations of TAC, 4.73% for high concentrations, −5.67% and 10% for MPA, and −4.65% and 1.75% for Cre. The data are included in Appendix A. No significant deviations (<±15%) were observed when comparing the EDTA-spiked samples with those without EDTA-spiked samples.

#### 3.2.8. Stability

After 24 h in the autosampler, the LQC samples of three analytes showed the largest mean deviations of 5.00%, 8.00%, and −7.92%, and the mean deviations of MQC and HQC were less than 2.50%. The maximum mean deviations at 4 °C and 60 °C (day 3) were −9.10% and −7.50% for TAC, 9.20% and −2% for MPA, and 5.19% and 4.25% for Cre. At 4 °C, 23 °C, 37 °C, and 60 °C (day 7) showed maximum mean deviations of −8.53%, 10.80%, −13.49%, and −21% for TAC; 11%, −7.40%, −8.90%, and −16.91% for MPA; for CRE, the values were −9.60%, −7%, −11.55%, and −18.27%. TAC showed a maximum mean deviation of −26.48% at 4 °C (one month) and −18.62% at 23 °C, while MPA showed −10.90% and −20.26%. Cre showed better stability at 4 °C and 23 °C (average deviation <±15%). All analytes exhibited good long-term stability (one month, mean deviation <±15%) at −20 °C, as shown in Appendix A.

### 3.3. Clinical Validation

Forty matched venous and capillary blood samples were collected from kidney transplant recipients for comparison and analysis. Five samples with MPA measured levels below the LLOQ in either capillary blood or plasma were excluded from method comparison statistics. Samples above the upper limit were diluted according to a validated dilution protocol. The basic conditions of the transplant recipients and the analysis results measured by different methods are shown in Appendix A and data analysis and results during clinical validation of three analytes are shown in Appendix A.

#### 3.3.1. TAC in VAMS and DBS

Compared with the TAC concentration range of 5.80 (4.97–7.25) ng/mL measured in venous blood, lower values were shown in both VAMS and DBS, which were 5.15 (4.47–6.49) ng/mL and 5.17 (4.32–6.19) ng/mL. No significant differences between methods were found when using PB regression analysis (Figure 3a,b). MPPE and MAPE were both <15% for TAC in VAMS and DBS, and ICC (0.93 and 0.88) showed a good correlation between methods. When applying Bland–Altman plots to study method agreement (Figure 3c,d), no significant deviations were detected (1.80% and 6.20%). The percentage of samples within acceptable limits of APE was 95.00% and 82.50% in VAMS and DBS, and 90.00% and 77.50% of samples met the criteria within clinically acceptable limits. No effect of hematocrit on the analysis was found (*p* = 0.95 and *p* = 0.97).

#### 3.3.2. MPA in VAMS and DBS

For mycophenolic acid, plasma concentrations were shown to range from 3.23 (2.52 to 4.15) µg/mL, significantly higher than those in VAMS and DBS between 1.83 (1.27–2.31) µg/mL and 1.65 (1.25–2.27) µg/mL, and the equation of the PB regression showed a very strong linear relationship, while the fitting performance indicators MPPE and MAPE were both lower than 15%. The regression equation can be used for concentration conversion. The formula for plasma concentration after the VAMS conversion was 0.04 + 1.87 × VAMS concentration, and the formula for the plasma concentration after the DBS conversion was 0.34 + 1.69 × DBS concentration. The PB regression between the corrected VAMS and DBS plasma concentrations and venous plasma concentrations found no significant differences (Figure 4a,b). Recalculated MPPE and MAPE were both <15% and the ICC (0.97 and 0.93) showed an excellent correlation between methods. When applying the Bland–Altman analysis (Figure 4c,d), no significant deviations were found (−1.50% for VAMS and 1.30% for DBS). In VAMS, the percentage of the paired samples within the limits of APE and the clinically acceptable range was 88.57% and 71.43%, and in DBS, it was relatively low (71.43% and 62.86%). When assessing the effect of the hematocrit, no significant differences were found between methods (*p* = 0.73 and *p* = 0.12).

#### 3.3.3. Cre in VAMS and DBS

Similar to mycophenolic acid, the creatinine serum concentration range was 115.50 ± 33.03 nmol/mL, which was higher than that in VAMS and DBS, 61.14 ± 22.93 nmol/mL in VAMS, and 65.47 (49.75–83.56) nmol/mL in DBS. Since both MPPE and MAPE were <15%, the correction equation generated by the PB regression analysis was used for concentration conversion. The VAMS creatinine correction formula was 24.25 + 1.46 × VAMS concentration, and the DBS creatinine was 32.19 + 1.21 × DBS concentration. PB regressions between corrected VAMS and DBS serum concentrations and venous serum showed good agreement (Figure 5a,b). The values of MPPE and MAPE were both <10%, and the ICC was 0.92 and 0.84 in VAMS and DBS. No significant deviations (−1.50% and 2.90%) were observed in the Bland–Altman analysis (Figure 5c,d). The percentage of paired samples meeting the APE and clinical constraints were higher in VAMS (92.50% and 85.00%) than in DBS (80.00% and 70.00%). No correlation was found between the hematocrit and concentration differences (*p* = 0.79 and *p* = 0.86).

### 3.4. Questionnaires

After clinical application, 40 questionnaires were collected to investigate patient satisfaction (20 for VAMS and 20 for DBS), as shown in Figure 6. The patient’s score for VAMS was 78.75 ± 10.52, and the score for DBS was 68.75 (63.75–75). Patient satisfaction with VAMS was higher than with DBS (*p* = 0.01).

## 4. Discussion

This study successfully developed and validated a VAMS-based UPLC-MS/MS method for the simultaneous quantification of TAC, MPA, and Cre in whole blood, which provides technical support for the transformation of the way of medical care seeks renal transplant recipients. Different from the previous report [16], our study extracted a shared VAMS sample, which required only a single injection without changing the analytical column and mobile phase to complete the quantification of three analytes simultaneously. This VAMS-based method is also the first to be reported to simultaneously perform immunosuppressants TDM and renal function monitoring in whole blood with only one sample injection and is of positive significance in alleviating the pain of patients and improving the work efficiency of laboratory personnel. Compared with other studies [26,27], the extraction steps and analysis process are more concise and efficient. It is undeniable that the mode transformation realized by VAMS can make the medical model more flexible, and the newly developed method not only facilitates users but also laboratory operators, which is extremely innovative and applicable among renal transplant recipients. We believe that this is a desire of both the recipient and the doctor. We chose venous whole blood for the construction of the standard curve to simulate the matrix of the VAMS application, which contained the irreducible endogenous creatinine. Therefore, the lower limit of quantification for Cre was evaluated during the validation process to ensure that the subsequent detection sample concentration can fall within the established standard curves.

During the establishment of the standard curve, the equilibration time of the standard added to the whole blood was compared. All analytes equilibrated for different times with acceptable deviations, and the maximum mean deviations for the three analytes (7.04%, −8.90%, and 6.14%) were minimal (<±10%) after one hour of equilibration, which was different from previous reports [26,27,28], but consistent with the study by Mathew [21]. It has been also reported that the presence of anticoagulants did not affect the quantification of tacrolimus and creatinine, and this was confirmed by the addition of an additional mycophenolic acid calibrator in our study. Evaluation of the relationship between hematocrit and quantification showed that the LC-MS/MS analysis of the VAMS samples was not affected by HCT, consistent with previous studies [16,17,21,26,28]. It has been previously reported that no significant matrix effect was observed in VAMS [21,27,29], which was also validated in our study. This was in contrast to the significant matrix inhibition (−34%) reported by Marshall et al. [16]. By comparison, our study is more suitable for patients and has good and consistent performance.

Dried VAMS samples were stable for more than seven days at ambient temperature, and they were also stable for at least three days when simulating extreme transport conditions (4 °C and 60 °C). So, it is recommended that the sample transportation time in high temperatures should not exceed three days to ensure the quality of the sample. Taking into account both home sampling needs and laboratory applications, long-term stability tests have been carried out. After one month of storage, a deviation of more than 15% was found in the analysis of TAC (at 4 °C and 23 °C) and MPA (at 23 °C), which was different from the existing reports [27,30]. The difference between the reports makes the long-term stability of the analytes at ambient temperature uncertain, but it was determined that TAC, MPA, and Cre were stable for one month at −20 °C, consistent with the previous reports [16,27,28]. This consistent result provides a sample storage solution for VAMS users who fail to mail on time and for laboratories that cannot immediately analyze received VAMS samples.

Our clinical validation in 40 kidney transplant recipients showed that TAC in VAMS and DBS was only slightly different from whole blood concentrations, both lower than WB concentrations, which was consistent with previous reports [9,17,30]. This also proves that the method we have established is effective. As for the reason, some scholars proposed that the anticoagulant contained in the WB sample does not exist in the capillary blood, or the incomplete filling in the VAMS sample [31,32]. Based on the good interchangeability between methods, we believe that methodological differences between immunoassays and mass spectrometry are the main reason for the lower concentrations in DBS and VAMS than those measured in whole blood. Whether additional reasons are related to manual spiking behavior and sample origin (fingertip blood may be diluted by tissue fluid) remains to be determined. Unlike the report of Paniagua and Mathew [17,21], we found more samples in VAMS to fall within the APE limits and set clinical limits (95.00% and 90.00%) relative to DBS (82.50% and 77.50%). The reason for the difference may be related to their rich experience in the DBS application, but it cannot be denied that the detection analysis of VAMS in TAC is better than that of DBS in our study.

For MPA, as expected (the presence of red blood cells dilutes MPA in plasma), MPA concentrations in VAMS and DBS were significantly lower than plasma MPA concentrations, which was similar to previous reports [21,30]. Although concentration correction using individual hematocrit was the most ideal way [33,34], considering that HCT cannot be accurately estimated when applying microsampling, our validation uses the PB regression equation for correction. Only two previous studies have clinically validated MPA in VAMS, reporting that 75.00% and 87.50% of the samples were within the set deviation (±20%) [17,30]. Our study showed that 88.57% of the samples were within acceptable APE limits, which was relatively consistent with that reported by Zwart (87.50%) [30]. For MPA in DBS, it showed lower concordance (71.43%) within the APE acceptance limit (±20%), which was consistent with previous reports [9,17,30,35]. Different from the previous study (reinjection analysis), the MPA concentration in our VAMS sample was detected by a single injection, and the results also showed good consistency, confirming the innovation and availability of the method. For the set clinical limits (±15%), the MPA levels obtained in both VAMS and DBS differ considerably from plasma, but this does not mean that the method we have established is not suitable for clinical application. It has been suggested to perform TDM on MPA levels in renal transplantation by using different analytical methods to optimize outcomes by reducing rejection or drug-related toxicity [36]. We believe that establishing a mass spectrometry reference range for MPA in whole blood may be more helpful to clinicians in individualized diagnosis and treatment.

Concentration levels of Cre in VAMS and DBS showed a constant proportional deviation before PB equation correction. Unlike previously reported proportions of creatinine bias (6.50% or 20–27%) [16,21,37], our study showed a systematic difference below 50% in serum creatinine concentrations. This difference may be caused by the fact that the enzyme method is used to detect the patient’s serum concentration while the VAMS method is used to detect the patient’s whole blood. Nevertheless, corrected creatinine concentrations in VAMS showed high concordance with serum creatinine, covering 85% of the sample with a clinically acceptable limit of ±15%, but less than 70% in DBS. It can be seen that the analysis of creatinine in VAMS has obvious advantages, which was consistent with the study of Mathew et al. [21].

Overall, for the three analytes (TAC, MPA, and Cre), VAMS, which draws a specific volume of blood, has a better analytical performance advantage over DBS. If kidney transplant recipients are unable to go to the hospital, it is undoubtedly a more convenient way to monitor renal function at the same time as TDM monitoring.

In terms of user satisfaction, our SUS score showed that kidney transplant recipients were more satisfied with the VAMS device than the DBS, and the feasibility of the application was also proved. The questionnaire survey of the renal transplant population also makes our research more practical. Clinicians have shown great interest in this method, believing that it can relieve the pressure of medical treatment for renal transplant recipients to a certain extent and facilitate postoperative management.

Of course, this study has limitations. In terms of controlling the quality of samples, in addition to collecting VAMS and DBS samples according to the specifications, we only relied on technicians to visually check whether the collected samples were qualified. This is of personal subjectivity, and the invisible unfilled cannot be observed. Moreover, it was found that the VAMS collected at the beginning was easy to fill, and the DBS collected later needed to squeeze the fingers moderately. Therefore, whether the collection order of VAMS samples and DBS samples will affect the quality of samples remains to be studied. It is required that the sample quality supervision should establish a recognized procedure to lay a foundation for future patient sampling training. Another limitation is the material cost of VAMS, which is more expensive than DBS. This should be considered in future clinical acceptance and application. At the same time, the questionnaires only included kidney transplant recipients who were willing to try it out, and we should collect more extensive opinions and ideas in the future. Before this technology is officially put into use, we should conduct more research, including the impact of the sampling quality of VAMS samples collected after user training on the accuracy of the results, the influence of real transport process rather than simulated transport conditions on the results, and the user-satisfaction of a wider range of renal transplant recipients. It is worth mentioning that whether the limitations of VAMS devices (high cost, single origin) can be broken in the future is also a blessing we hope to bring to renal transplant recipients. On the whole, this study provides essential technical support and some willingness surveys and is the steppingstone to realize the transformation of the remote monitoring mode, which has profound significance. In the future, more details need to be further studied to facilitate renal transplant recipients.

## 5. Conclusions

A stable method for the simultaneous quantification of tacrolimus, mycophenolic acid, and creatinine concentrations in VAMS whole-blood samples was initially established and fully verified. This method provides analytical support for the one-sample remote monitoring of both immunosuppressive drug concentrations and renal function in allo-renal recipients. Based on the better analytical performance and willingness survey of this method than DBS after application in the study, we believe that VAMS sampling can be a superior alternative to venous whole-blood sampling.

## Figures and Tables

**Figure 1 pharmaceutics-14-02547-f001:**
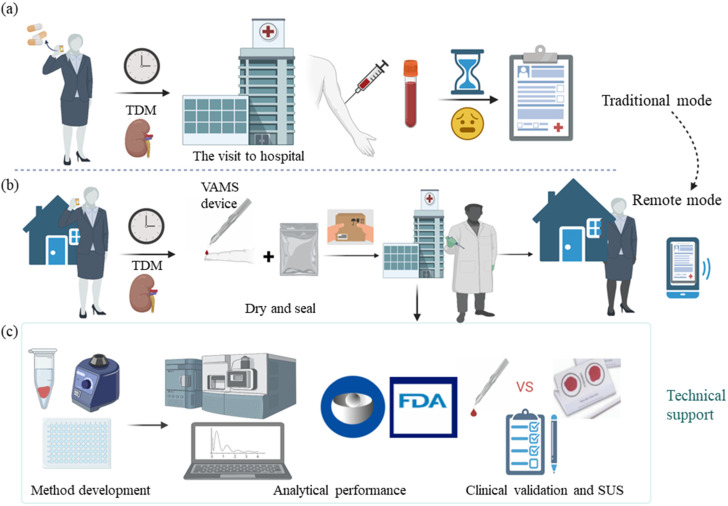
Schematic diagram of the research content. (**a**) the traditional mode of TDM and renal function monitoring in renal transplant recipients. They must go to the hospital regularly for venous blood sampling and wait hours to get their reports. It must be admitted that the visit to the hospital is time-, labor-, and cost-consuming. (**b**) It shows the desired remote monitoring mode if a VAMS device is applied. In this way, the recipients can conduct microsampling of fingertip blood at home, and then the VAMS filled with capillary blood will be dried, sealed, and mailed to the hospital for analysis. This mode will allow them to receive an electronic report at home and complete regular TDM and renal function monitoring. (**c**) We developed and established a new robust method for simultaneous monitoring of TDM and renal function using single-sample analysis to provide technical support. The research contents include method development, analytical performance, clinical validation (compare DBS and VAMS), and a questionnaire (SUS) to assess feasibility. Materials sourced from BioRender.com.

**Figure 2 pharmaceutics-14-02547-f002:**
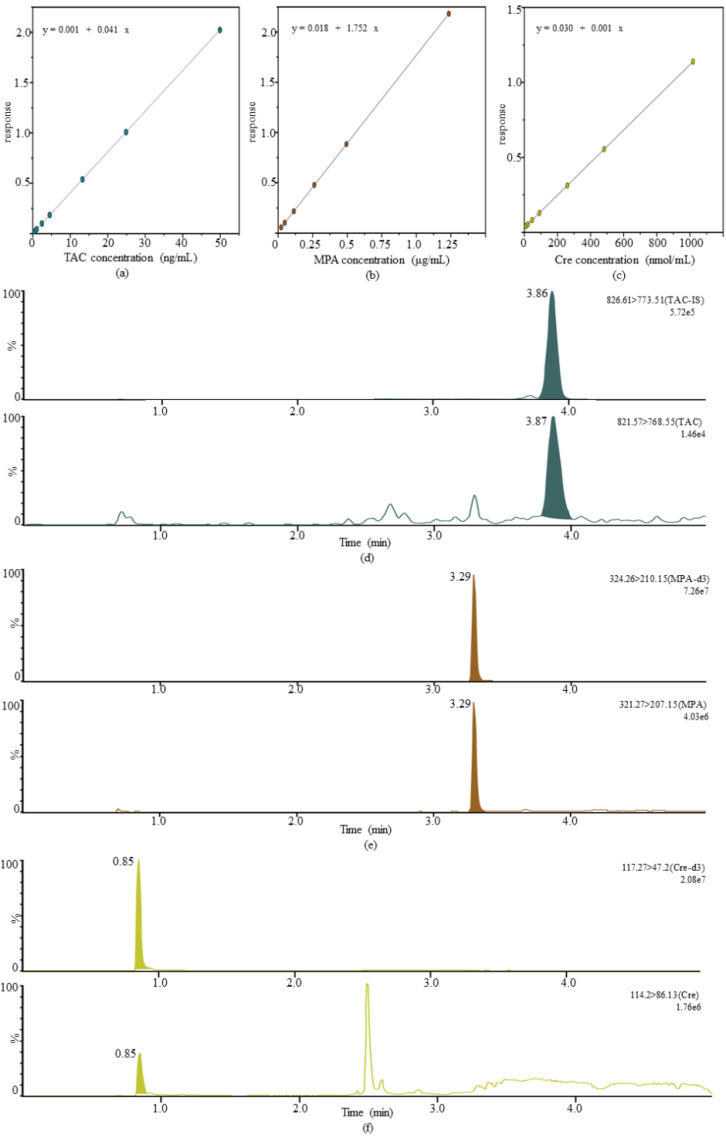
Standard curves and LLOQ ion chromatogram peaks for the three analytes. (**a**) The standard curve range of TAC is 0.50~50.00 ng/mL; (**b**) The standard curve range of MPA is 0.025~1.25 μg/mL; (**c**) The standard curve range of Cre is 10.00~1000.00 nmol/mL; (**d**) The internal standard channel and analyte channel of tacrolimus LLOQ (0.50 ng/mL) of channel; (**e**) The internal standard channel and analyte channel of mycophenolic acid LLOQ (0.025 μg/mL); (**f**) The internal standard channel and analyte channel of creatinine LLOQ (10.00 nmol/mL).

**Figure 3 pharmaceutics-14-02547-f003:**
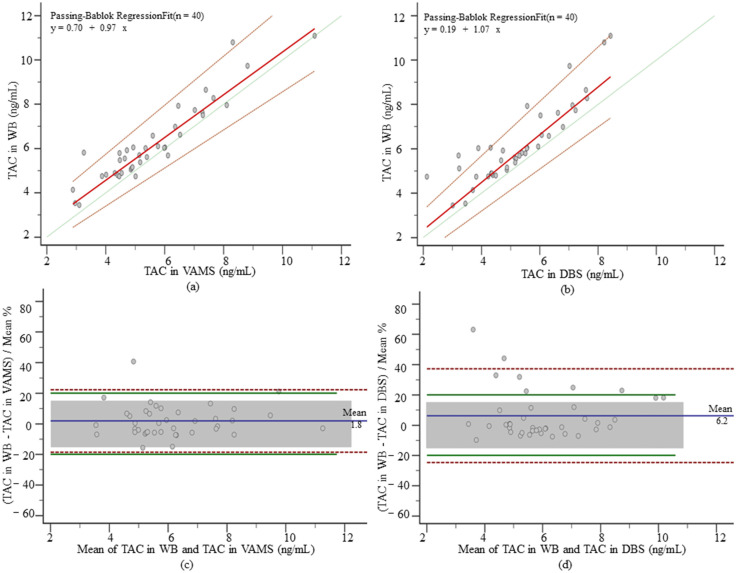
PB regression and Bland–Altman plot for TAC. (**a**) PB regression of TAC in VAMS samples and whole-blood samples, the solid red line is the regression line, the dashed red line is the 95% CI, and the solid green line is the identity; (**b**) PB regression of TAC in DBS samples and whole-blood samples; (**c**) Bland–Altman plot of whole blood TAC concentration and TAC concentration in VAMS. The blue solid line is the mean deviation, the red dashed line is the 95% CI, the green solid line is the APE acceptance limit (±20%), and the grey box is the set clinical acceptance limit (±15%), the following MPA and Cre as well; (**d**) Bland–Altman plot of whole blood TAC concentration and TAC concentration in DBS.

**Figure 4 pharmaceutics-14-02547-f004:**
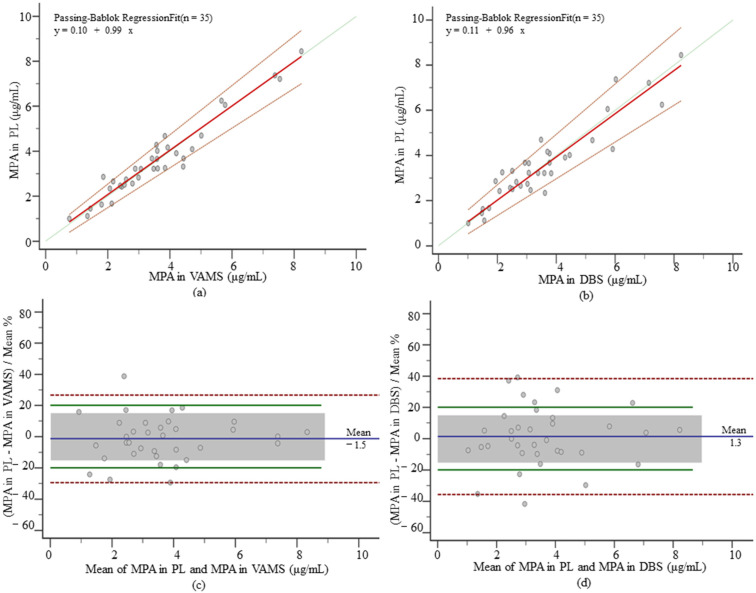
PB regression and Bland–Altman plot for MPA. (**a**) PB regression of corrected MPA concentrations in VAMS samples and plasma samples, the solid red line is the regression line, the dashed red line is the 95% CI, and the solid green line is the identity; (**b**) PB regression of corrected MPA concentrations in DBS samples and plasma samples; (**c**) Bland–Altman plots of plasma MPA concentrations and corrected VAMS plasma concentrations. The blue solid line is the mean deviation, the red dashed line is the 95% CI, the green solid line is the APE acceptance limit (±20%), and the grey box is the set clinical acceptance limit (±15%), the following MPA and Cre as well; (**d**) Bland–Altman plots of plasma MPA concentrations and corrected DBS plasma concentrations.

**Figure 5 pharmaceutics-14-02547-f005:**
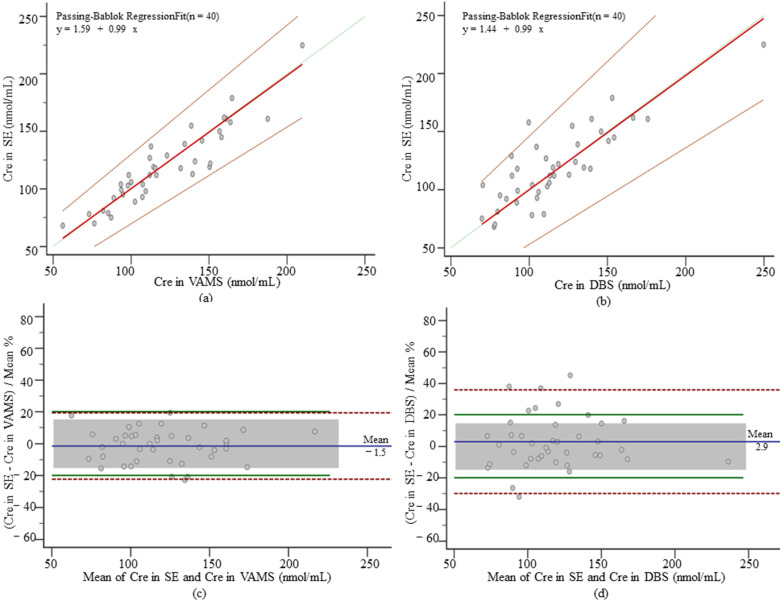
PB regression and Bland–Altman plot for Cre. (**a**) PB regression of corrected Cre concentrations in VAMS samples and serum samples, the solid red line is the regression line, the dashed red line is the 95% CI, and the solid green line is the identity; (**b**) PB regression of corrected Cre concentrations in DBS samples and serum samples; (**c**) Bland–Altman plots of serum Cre concentrations and corrected VAMS serum concentrations. The blue solid line is the mean deviation, the red dashed line is the 95% CI, the green solid line is the APE acceptance limit (±20%), and the grey box is the set clinical acceptance limit (±15%), the following MPA and Cre as well; (**d**) Bland–Altman plots of serum Cre concentrations and corrected DBS serum concentrations.

**Figure 6 pharmaceutics-14-02547-f006:**
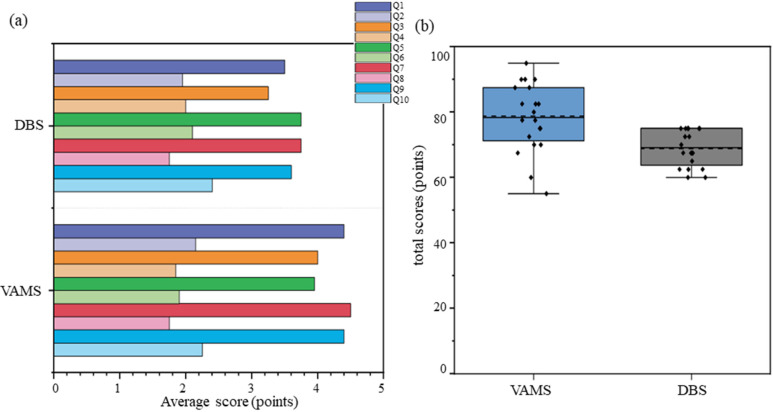
Average score and total scores of the SUS from VAMS and DBS. (**a**) The question for Q1 is: I think that I would like to use this system frequently. The question for Q2 is: I found the system unnecessarily complex. The question for Q3 is: I thought the system was easy to use. The question for Q4 is: I think that I would need the support of a technical person to be able to use this system. The question for Q5 is: I found the various functions in this system were well integrated. The question for Q6 is: I thought there was too much inconsistency in this system. The question for Q7 is: I would imagine that most people would learn to use this system very quickly. The question for Q8 is: I found the system very cumbersome to use. The question for Q9 is: I felt very confident using the system. The question for Q10 is: I needed to learn a lot of things before I could get going with this system. (**b**) Boxplots of SUS total scores for VAMS and DBS. The dashed black line is the median line, and the solid black line is the mean line.

**Table 1 pharmaceutics-14-02547-t001:** Mass spectrometer settings for analytes.

Analyte	Precursor Ion (*m*/*z*)	Production (*m*/*z*)	Cone (V)	Collision (V)	Retention Time
TAC	821.57	768.55	24	18	3.87
TAC [13C,2H4]	826.61	773.51	14	18	3.87
MPA	321.27	207.15	10	22	3.29
MPA-d3	324.26	210.15	8	22	3.29
Cre	114.20	86.13	44	10	0.85
Cre-d3	117.27	47.20	38	14	0.85

**Table 2 pharmaceutics-14-02547-t002:** Within- and between-batch accuracy and precision (N = 5).

		Day 1		Day 2		Day 3		Interbatch CV (%)
		Mean Deviation (%)	CV (%)	Mean Deviation (%)	CV (%)	Mean Deviation (%)	CV (%)	
TAC	LLOQ (0.50)	−6.64	7.74	0.68	5.30	−3.12	5.24	6.52
(ng/mL)	LQC (1.50)	1.79	5.18	0.94	4.39	2.71	2.77	3.98
	MQC (20.00)	9.10	2.47	10.52	2.52	11.87	2.03	2.42
	HQC (40.00)	5.38	4.53	8.76	4.27	9.46	3.40	4.15
MPA	LLOQ (0.025)	−8.00	4.35	−11.20	2.01	−11.20	3.96	3.66
(μg/mL)	LQC (0.075)	−0.64	3.18	−4.33	4.19	−5.52	1.82	3.72
	MQC (0.50)	−7.16	4.12	−8.00	4.95	−8.92	5.50	4.59
	HQC (1.00)	−11.54	1.42	−12.56	1.97	−12.66	1.10	1.55
CRE	LLOQ (10.00)	7.27	5.33	6.02	4.80	1.51	5.57	5.42
(nmol/mL)	LQC (30.00)	1.68	5.21	5.28	6.86	−1.17	3.08	5.64
	MQC (400.00)	−0.54	5.30	−1.46	4.59	−2.15	4.53	4.52
	HQC (800.00)	−3.26	2.82	−6.07	3.19	−7.02	3.03	3.29

**Table 3 pharmaceutics-14-02547-t003:** Extraction recovery and process efficiency.

	Recovery Rate (%)			Process Efficiency (%)		
Analytes	LQC	MQC	HQC	LQC	MQC	HQC
TAC	96.49	105.89	99.21	104.43	109.28	101.42
MPA	97.13	95.01	100.20	96.08	99.11	98.08
CRE	101.75	95.09	96.26	96.65	95.88	98.62

## Data Availability

Data are contained within the article.

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
