# Peer review of "A Volumetric Absorptive Microsampling UPLC-MS/MS Method for Simultaneous Quantification of Tacrolimus, Mycophenolic Acid and Creatinine in Whole Blood of Renal Transplant Recipients"

_pharmaceutics, 2022, doi:10.3390/pharmaceutics14122547_

Round 1
Reviewer 1 Report
To the authors:
This is an interesting paper. Authors provide new method for the simultaneous determination of tacrolimus, mycophenolic acide and serum creatinine. In order to meet wider audience, some answers need to be provided.
Abstract
“Development and validation were carried out following the guidelines for the validation of bioanalytical methods” Which guideline? It should be named.
“VAMS sampling is a superior alternative to venous blood sampling compared to DBS”. This, last sentence in the abstract should be clearer for the reader. Rephrase this part.
Introduction
Please your comment on TDM, that it is not only concentration determination, but results interpretation. Who will do that? Most patients have visited their doctor in that same time they have TDM sampling at the Clinic.
Results
Why did not you compare VAMS and venous blood samples with LC-MS/MS for all three parameters?
Discussion
Limitations of the study, strengths and weaknesses, future perspectives are needed.
Conclusion
It should be elaborated with more details
English should be improved
Author Response
请参阅附件。

Reviewer 2 Report
Manuscript pharmaceutics-1950652 entitled “A volumetric absorptive micro sampling UPLC-MS/MS method for simultaneous quantification of tacrolimus, mycophenolic acid and creatinine in whole blood of renal transplant recipients” developed a bioassay to quantify drugs and a biomarker, creatinine at the same time. In general, the manuscript is well-written. However, the following comments should be addressed before consideration:
1. Section 2.1., please state the purity of standard compounds used.
2. Section 2.2., “The gradient elution program was set to 2% B for 1.0 min, then the percentage of B was increased to 50% B in 0.5 min, then to 90% B at 2.5 min, and held 90% B for 2.7 minutes. At 4.2 minutes, set B to 2%. The total injector run time was 5 minutes and the injection volume for a single analysis was 2μL” The description of the timepoints in gradient elution is very confused. According to the statement, it is assumed B phase at 90% was hold for 1.7 minutes, not 2.7 minutes. Please correct the error or re-write the sentences if the assumption is wrong.
3. Section 2.3., QCL concentrations for TAC, MPA and Cre were 4X, 2X and 4X of their corresponding LLOQ, which the setting was not obeyed FDA guideline (i.e., low QC = 3 times of LLOQ). Please explain it.
4. Section 2.4., please use gravity force instead of rpm to state the vortex parameter.
5. In this manuscript, the authors used the authentic matrix (i.e. whole blood) to prepare the calibrators and QCs for validating the creatinine method. Since the whole blood is creatinine-contained, the endogenous creatinine level cannot be ignored. By comparing the creatinine chromatographs of untreated sample and LLOQ sample, the significant peak representing Creatinine is found in the untreated samples. It impacts the sensitivity and specificity of the method. Although the authors stated all measured results of creatinine in the samples could fall within the calibration range, the lot-to-lot difference of creatinine level between the blank matrix for preparing calibrators and clinical samples would also impact the accuracy and precision of the measured results, especially at the lower end. A typical way of validating a compound circulates in blood is to find a compound free matrix and do a parallelism study. Please make a good explanation.
6. Round all numbers to the same decimal places through the manuscript.
Reviewer 3 Report
The authors have developed a method based on volumetric absorptive microsampling (VAMS) combined with Ultra Performance Liquid Chromatography-Tandem Mass Spectrometry (UPLC MS/MS) to simultaneously quantify TAC, MPA, and Cre in whole blood. Moreover, they have presented the validation as well the clinical validation. Finally, a survey was conducted with renal transplant recipients.
It is an interesting work, but there are some points to be either corrected or inserted:
i) I recommend inserting a scheme to represent the volumetric absorptive microsampling (VAMS). It is a core topic of this manuscript;
ii) In subitem "2.4 Shared sample extraction", the authors have mentioned that the samples were prepared according to the instructions. Please detail these instructions.
iii) In item "3. Results", the authors have inserted a small paragraph to describe the concentration measured by sampling at different equilibration times. In my opinion, it lacks a subtitle to describe these results. Additionally, I recommend detailing this procedure.
iv) I recommend presenting the equation, intercept and slope for the calibration curves presented. Moreover, as presented in Figure 2a, I noticed that the authors have plotted the mean of the replicates for each concentration of the calibration curve. Why the authors have chosen the mean instead of plotting the points separately?
v) In Table S2, the column titles were repeated: "95% CI intercept by PB regression". I think that one refers to the intercept and the other to the slope.
vi) In the same columns of Table S2, please check the values for "DBS vs PL" and "corrected DBS vs PL", it is different from the values mentioned in the manuscript.
Reviewer 4 Report
The manuscript describes a new analitycal method for tacrolimus and mycophenolic acid by capillary microsampling by means of VAMS. Method is well described and data are consistent. Of note the possibility to determine also plasmatic creatinine with TAC and MPA in kidney transplated patients.
Manuscript requires some english language/word/style check: After that it may be published in its present form.
effetively the manuscript seems to me to be a good manuscript as I have already mentioned. It is true that using creatinine as a normalizing factor may pose a problem being an endogenous molecule. However, the authors point out that they are aware that this can be a problem and so I expect them to make their assessments on each new individual matrix pool to construct the calibration curves and QCs. Definitely tedious work but of which the authors seem to be aware. Having said that I reiterate my previous assessment and believe that the work can be published and is certainly of interest to Pharmaceutics readers.
Reviewer 5 Report
The authors reports a volumetric absorptive micro sampling (VAMS) UPLC-MS/MS method for simultaneous quantification of tacrolimus, mycophenolic acid and creatinine in whole blood of renal transplant recipients. The authors also report an interesting comparison with the DBS sampling method.
The article is well organized and written
Some suggestions:
1. Please reduce the validation paragraph. Describe briefly the parameters eliminating lengthy descriptions.
2. Emphasize better the novelty and limitations of the discovery even in comparison to the previous literature.
3. Add an idea of what could or should still be done in relation to the issue addressed in this paper. The authors should synthesize the eventual applications and the possibilities in the field of renal transplant drug monitoring.
4. minor text editing is required. e.g. use the same measurement unit in the text (ng/mL or ug/mL,...)
Round 2
Reviewer 2 Report
The authors addressed all my questions clearly.
Author Response
Thank you for your valuable time.